# DYNAMIC INFORMATION SUB-SELECTION FOR DECISION SUPPORT

## ABSTRACT

We introduce Dynamic Information Sub-Selection (DISS), a novel framework of AI assistance designed to enhance the performance of black-box decision-makers by tailoring their information processing on a per-instance basis. Blackbox decision-makers (e.g. humans or real-time systems) often face challenges in processing all possible information at hand (e.g. due to cognitive biases or resource constraints), which can degrade decision efficacy. DISS addresses these challenges through a policy that dynamically select the most effective features and options to forward to the black-box decision-maker for prediction. We develop a scalable frequentist data acquisition strategy and a decision-maker mimicking technique for enhanced budget efficiency. We explore several impactful applications of DISS, including biased decision-maker support, expert assignment optimization, large language model decision support, and interpretability. Empirical validation of our proposed DISS methodology shows superior performance to state-of-the-art methods across various applications.

## 1 INTRODUCTION

Typical machine learning (ML) approaches process all available information across all instances. However, computational costs, biases, and other limitations of decision-makers often make it so that they are better served by a per-instance *dynamic* adjustment of what and how information are processed when making decisions. For example, human decision-makers may actually do worse with more information on an instance due to cognitive overload (Iskander, 2018), thereby necessitating a concise summary of the most relevant features per instance (Morris, 2018). Moreover, ML on edge devices, such as microcontrollers, may be subject to strict time and memory constraints (Plastiras et al., 2018), making it so that it is only possible to make inferences with a subset of features. In this work, we introduce *a new class of AI assistance* for black-box decision makers (BDMs) based on dynamic information sub-selection (DISS)

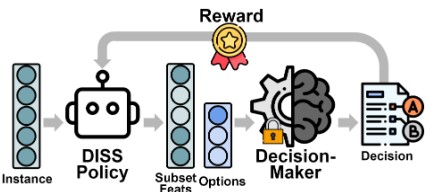

Figure 1: Given an instance, DISS policy selects and forwards a feature subset (and option) to a decision-maker to render its decisions. A reward evaluating the quality of the decision maker's decision is used as feedback to DISS policy.

that determines what pieces of information to pass along to a BDM in a per-instance-personalized fashion; we develop novel methods to learn policies that tune what information to utilize when making decisions with BDMs.

**Approach** A core principle of this work is a data-driven approach that directly considers the efficacy of the BDM when given various potential subsets of features (and other options) for prediction on instances. Therefore, we pose DISS as a reward maximization problem: given an instance, the DISS policy selects information and options, passes its selection to the BDM, and obtains a reward according to the efficacy of the decision made by BDM (e.g. accuracy, confidence, time, etc.); see Fig. 1 for illustration. Therefore, obtaining decision-making observations directly from the BDM is vital to tuning the DISS policy. However, in many applications, there are limitations to how many decisions one can query from the BDM during training due to restrictions in computational resources, time, monetary incentives, etc. Hence, this paper develops an acquisition strategy to ef-

fectively learn a policy under budgeted observational constraints that judiciously chooses the next query based on past collected observations from the BDM. Moreover, we propose a new regressor and show that we may make even better use of our observations by *exploiting the structure of our problem* by learning to mimic the BDM.

**Applications** DISS offers a range of real-world applications by supporting decision-making processes involving general black box decision-makers, including (but not limited to) the following. *1) Human decision support*. DISS can help mitigate human cognitive limitations and biases (Griffiths, 2020; Buschman et al., 2011; Caruana et al., 2020) by tailoring what and how instance information gets presented to human decision makers as they make decisions (e.g. on patient cases). *2) Multi-expert selection*. In real-world scenario, one might have access multiple experts (e.g. statistical models) each with its own specialties; in this case, DISS can be used to not only customize what and how information presented but also what expert to employ on a current given instance. *3) LLM decision support*. Recent success in large language model (LLM) has drawn huge interest from users across all disciplines (W. X. Zhao et al., 2023); however, prior works have shown that LLMs can be sensitive to prompt styles (or how one asks questions to LLM) (Liu et al., 2024; Z. Zhao et al., 2021); DISS can enhance the quality of decisions made by LLMs via choosing between prompt styles and selecting what information to forward to an LLM. *4) Interpretability and Visualization*. By selecting a dynamic (per-instance) small subset of the most relevant features, DISS presents a complementary approach in feature importance and interpretability (e.g. Saarela & Jauhiainen (2021), Yongjie Wang et al. (2024), and Lakkaraju et al. (2017)) to construct interpretable ML models that can be visualized easily whilst producing efficacious decisions.

**Contributions** Our contributions are: 1) present a simple to implement bootstrap approach enabling the use of arbitrary regressors to choose context-action pairs for learning DISS policies; 2) further introduce and analyze a decision-maker mimicking approach that better utilizes available data; 3) frame multiple real-world applications within our proposed DISS framework; 4) empirically show efficacy of our methods against state-of-the-art methods under various settings and datasets.

## 2 RELATED WORK

**Contextual Bandits** DISS may be framed as a contextual multi-armed bandit problem (Lu et al., 2010), where the context is an instance $x \in \mathbb{R}^d$, the action space include all possible subsets of $d$ features (and a finite set of options), and rewards stem from the BDM's prediction quality when given the subset of features (and options) from the context (see Lattimore & Szepesvári (2020) for a survey on contextual bandits). Note that the cardinality of action space $|\mathcal{A}| \geq 2^d$ grows exponentially in the number of features, making it difficult for contextual bandits; this in itself is an active research area termed combinatorial multi-armed bandit (Gai et al., 2012; W. Chen, Yajun Wang & Yuan, 2013; W. Chen, Yajun Wang, Yuan & Q. Wang, 2016; S. Wang & W. Chen, 2018; Zierahn et al., 2023); yet, to the best of our knowledge, existing works in combinatorial multi-armed bandit mostly consider a special type of bandit termed semi-bandit in which the rewards of pulling individual base arm are also revealed when a super arm is chosen[1]; however, rewards are not additive over features for our DISS scenario and there are no individual arm rewards, making such approach incompatible with our setting. Zhu et al. (2022) proposed SpannerIGW supporting contextual bandit with large action space; however, it does not cache prior observations and takes more interactions to see improvement in rewards; as shown in Appx. E, SpannerIGW needs notably more budget to achieve comparable reward in DISS setting.

**Adaptable Displays** The problem of providing decision support via adaptive interfaces has been studied in the context of human-computer interaction (HCI). W. Wang et al. (2024) surveyed adaptive user interfaces for long-term health patients with predictive algorithms (e.g. reinforcement learning, Monte Carlo tree search). Typically, previous approaches consider a limited number of options to adjust interfaces for human decision-making. For example, Buçinca et al. (2024) explored offline reinforcement learning to learn policies that adjust four options (vary the type of AI prediction assistance to provide) to aid human learning of tasks. Lomas et al. (2016) employed multi-armed bandits to optimize the experience of an online educational game by selecting among

---

[1]Super arms in combinatorial multi-armed bandit are sets of arms with its elements being individual "base" arms; in DISS, base arms and super arms correspond to choosing individual features and choosing feature subsets, respectively.

six interface variants. Modiste (U. Bhatt et al., 2023) employed KNN-UCB (Guan & Jiang, 2018), a variant of UCB, to learn a policy used to assist human decision-maker's accuracy through providing personalized additional relevant information (over three options) such as LLM output. DISS, in contrast, selects over a much larger number of adjustments than prior approaches, and also covers far more applications beyond human-assistance.

**LLM Prompt Engineering**    LLMs can adapt to a myriad of tasks with adequate prompting (Radford et al., 2019). Recent works have studied various prompting heuristics, including in-context learning (Brown et al., 2020), chain-of-thought prompting (Wei et al., 2022) and prompt-tuning (Lester et al., 2021). However, improper prompting may cause LLMs to output poor responses (Burnell et al., 2023). Moreover, recent works also show that the LLM outputs are sensitive to the prompting style, position of information in the prompt, and the length of the prompt (Liu et al., 2024; Z. Zhao et al., 2021; Anagnostidis & Bulian, 2024). Zollo et al. (2023) introduces an approach to select prompts from a static set based on statistical upper bounds to avoid unacceptable outcomes generated by LLMs. However, this line of work is limited to the expertise of the prompt designer to create a relatively small fixed set of candidate prompts. Instead, our approach presents an alternate formulation that encompasses such prompting style choices along with a per-instance tailored curation of information to forward to an LLM.

# 3 METHODS

## 3.1 PROBLEM SETTING

**Decision Maker**    We seek to maximize rewards for the decisions of a BDM $M$ that ingests: 1) an arbitrary subset of available information on an instance $x$; 2) an additional option $o$ (both of which will impact $M$'s output). To illustrate, a clinician may ingest a given subset of patient features in a visual display style specified by $o$. We take $x \in \mathbb{R}^d$ for notational simplicity. For each instance, we consider $M$'s decision space to be binary for simplicity, $\widehat{y} \in \{0, 1\}$ (e.g. whether to diagnose a patient with a certain disease). In this case, the BDM $M$ makes decisions based on a subset of features $b \in \{0, 1\}^d$ and an additional option $o \in \mathbb{O}$ as $M(x \odot b, o)^2$, where $\odot$ denotes the Hadamard product. We consider the output space of $M$ to be $[0, 1] \times \mathbb{M}$, the tuple of $M$'s probability of selecting $\widehat{y} = 1$, along with any additional meta-data $m \in \mathbb{M}$ that is relevant for assessing decision quality. For example, a clinician outputs a confidence of their decision (probability of $\widehat{y} = 1$), along with a recorded time taken to make a decision (the meta-data $m$).

**DISS Policy and Criterion**    The policy shall receive a reward $r(y, M(x \odot b, o))$ based on the output of the decision maker $M$ and the ground truth optimal decision $y$ for an instance $x$. We take the reward function to be known, which reflects typical use cases where the output of the decision-maker, meta-data, and ground truth optimal decisions are deterministically aggregated in some known, application specific manner to measure decision efficacy (e.g. based on accuracy, confidence, speed to decision, etc.). We wish to learn a policy $\pi : \mathbb{R}^d \mapsto \{0, 1\}^d \times \mathbb{O}$, that maximizes this reward over a data distribution of instances, $\mathcal{D}$. That is, we want to learn a policy $\pi$ as:

$$\arg\max_{\pi \in \Pi} \mathbb{E}_{(x,y) \sim \mathcal{D}}[r(y, M(x \odot \pi_b(x), \pi_o(x)))], \tag{1}$$

where $\pi_b(x)$, $\pi_o(x)$ indicate the selected subset of features and options, respectively. To train the policy, we assume that we have a standard supervised dataset $D = \{(x^{(i)}, y^{(i)})\}_{i=1}^n \overset{iid}{\sim} \mathcal{D}$. Yet, note that $D$ does not contain any information about the decision-making of $M$, which the policy must consider. For many real-world applications, we will be limited on the number of calls that can be made to the BDM $M$ during training; for instance, time/incentive limitations on collecting decision-making data from clinicians. Thus, we consider a fixed-budget of $K$ queries to $M$ when training $\pi$. That is, we consider a paradigm where one sequentially queries the BDM $M$ at $K$ training instances, i.e. $M(x^{(i_1)} \odot b^{(1)}, o^{(1)}), \dots, M(x^{(i_K)} \odot b^{(K)}, o^{(K)})$, where $x^{(i_l)} \in D$ is the queried training instances, $b^{(l)} \in \{0, 1\}^d$ is the respective information subsets, and $o^{(l)} \in \mathbb{O}$ is the selected options; we similarly record rewards $r(y^{(i_1)}, M(x^{(i_1)} \odot b^{(1)}, o^{(1)})), \dots, r(y^{(i_K)}, M(x^{(i_K)} \odot b^{(K)}, o^{(K)}))$ corresponds to each query. Let $B^{(K)}$ be the dataset of the $K$ acquired decision-making observations:

$$B^{(K)} \equiv \{(i_l, b^{(l)}, o^{(l)}, \eta^{(l)}, \rho^{(l)})\}_{l=1}^K, \tag{2}$$

---

[2]Note that, depending on the domain of instance features, $x \odot b$ may be ambiguous between omitted and zero values. $M$ may equivalently be written as $M(x \odot b + \varnothing(\mathbf{1} - b), o)$ for an empty token $\varnothing$ or as $M(x \odot b, b, o)$. We omit this for notational simplicity.

where $\eta^{(l)} = M(x^{(i_l)} \odot b^{(l)}, o^{(l)})$ and $\rho^{(l)} = r(y^{(i_l)}, \eta^{(l)})$. We focus on test-time rewards (eq. 1) after an initial training phase where decisions are collected based on historical offline instances $D$ (see Char et al., 2019 for further details on offline multitask reward optimization).

## 3.2 MIMIC-STRUCTURED REGRESSION

Below, we propose to leverage a special structure to the DISS regression task and propose to decompose the estimation into two components that makes better use of the data at hand. Recall that in our DISS framework, the reward function $r$ measuring decision-making efficacy is known. We now study the estimated target of our regression task, $f(x, b, o) \approx \mathbb{E}_{y|x}[r(y, M(x \odot b, o))]$. Upon inspection, one may note that the target actually depends on two unknown quantities: 1) $p(y \mid x)$, the ground truth conditional likelihood for the optimal decision given $x$; and 2) $M(x \odot b, o)$ the output of the decision-maker at $x$ with a configuration $b, o$. This observation, which has not been previously used in contextual bandit decision support literature (to the best of our knowledge), enables us to better leverage our data. Namely, $p(y \mid x)$ *is independent of the blackbox decision-maker*, and can directly be estimated in a supervised fashion given our dataset of instances $D = \{(x^{(i)}, y^{(i)})\}_{i=1}^{n}$. That is, we propose to directly estimate $\widehat{p}(y \mid x)$ (a classifier in the discrete $y$ case), which avoids any budget expenditure on observations of $M$. On the other hand, $M(x \odot b, o)$ depends on the decision-maker and must be estimated with budgeted observations. We propose to learn to *mimic* the BDM based on the observations in $B^{(T)}$ (eq. 2) in a supervised fashion:

$$\widehat{M} \equiv \arg\min_{M'} \sum_{t=1}^{T} \ell(M'(x^{(i_t)} \odot b^{(t)}, o^{(t)}), \eta^{(t)}), \tag{3}$$

where $\ell$ is a supervised loss (e.g. cross-entropy), and $\eta^{(l)} = M(x^{(i_l)} \odot b^{(l)}, o^{(l)})$ are the observed decisions (and meta-data) from the BDM. With both $\widehat{p}(y \mid x)$ and $\widehat{M}$, we can directly estimate rewards using the mimic-structured regressor:

$$\widehat{f}_{\mathrm{MS}}(x, b, o) \equiv \mathbb{E}_{y \sim \widehat{p}(y|x)}[r(y, \widehat{M}(x \odot b, o))]. \tag{4}$$

In Appx. A we expound on a risk upper-bound and on the efficiency of the mimic-structured estimator, which essentially estimates two classifiers (one of which, $\widehat{p}(y \mid x)$ is fitted on un-budgeted data) versus the vanilla regression approach, which can be viewed as fitting either $2^d$ regressors in $d$ dimensions, or as fitting a regressor in a larger $2 \cdot d$ space.

**Proposition 3.1** (Risk Upper Bound, Informal). *Given regularity conditions, we show our proposed mimic-structured has a risk upper-bound of*

$$\mathbb{E}\left[\left|\mathbb{E}_{p(y|x)}\left[r(y, M(x \odot b))\right] - \mathbb{E}_{\widehat{p}(y|x)}\left[r(y, \widehat{M}(x \odot b))\right]\right|\right] \leq c'' t^{-\frac{1}{d+2}}$$

*where $t$ is the budgeted observations on the BDM and estimate $\widehat{M}$ is trained on an iid dataset $B = \{(x^{(j)} \odot b^{(j)}, \eta^{(j)})\}_{j=1}^{t}, \eta^{(j)} \sim M(x^{(j)} \odot b^{(j)})$.*

As expounded in Appx. A, we see a better error rate than the corresponding vanilla regressor over $x$ and $b$. Our experimental results (Fig. 2) show improved estimates/reward maximization extends even in non-*iid* adaptive acquisition strategies.

**Active Training Observation Acquisition** Recall that we are operating under a budget constraint for obtaining BDM decision-making observations (eq. 2). Typically acquisition strategies (e.g. Thompson sampling, see below) assume the ability to draw from a posterior regressor, which is not available for our mimic-structured regressor. To overcome this, we propose to utilize a frequentist Thompson sampling (FTS) routine (§ 3.3). We may use FTS (Fig. 1) with the mimic-sturctured regressor by estimating $\widehat{p}(y \mid x)$ and $\widehat{M}$ with respective bootstrap trials, which yields a natural way of acquiring data to resolve ambiguities between our knowledge of what the BDM will do and the impact on rewards.

## 3.3 FREQUENTIST THOMPSON SAMPLING ACQUISITION

**Bayesian Regressor-based Policy** Our task connects nicely to Bayesian experimental design and Bayesian Optimization (Greenhill et al., 2020) as we must decide the $K$ decision-making observations to acquire in order to train our policy. Thus, it is natural to consider a Gaussian Process

(GP, Ling et al., 2016) based reward estimator $f \sim \mathcal{GP}(\mu, \sigma)|B^{(K)}$, which estimates $f(x, b, o) \approx \mathbb{E}_{y|x}[r(y, M(x \odot b, o))]$, the expected reward on an instance $x$ with features and options $b, o$ on the BDM $M$ based on observations $B^{(K)}$ (eq. 2). Let $\mu^{(K)}(x, b, o) = \mathbb{E}_{f \sim \mathcal{GP}(\mu, \sigma)|B^{(K)}}[f(x, b, o)]$ be the corresponding posterior mean. Note that one may derive a policy based on $\mu^{(K)}$ as $\pi(x) \equiv \operatorname{argmax}_{b,o} \mu^{(K)}(x, b, o)$. Below, we describe strategies to acquire observational data for this policy.

**Thompson Sampling Acquisition** Thompson sampling (TS) (Thompson, 1933; Thompson, 1935; Russo et al., 2018) provides a principled approach to acquire an additional observation at step $t$ based on the previously acquired data $B^{(t-1)}$ (eq. 2). We propose using Thompson Sampling to collect the next observation tuple based on a uniformly sampled instance $i_t \sim \operatorname{Unif}\{1, \ldots, n\}$ and posterior draw $\widetilde{f} \sim \mathcal{GP}(\mu, \sigma)|B^{(t-1)}$ using

$$b^{(t)}, o^{(t)} \equiv \operatorname{argmax}_{b,o} \widetilde{f}(x^{(i_t)}, b, o) \tag{5}$$

to update the data as $B(t) \equiv \{(i_t, b^{(t)}, o^{(t)}, \eta^{(t)}, \rho^{(t)})\} \cup B^{(t-1)}$, where $\eta^{(t)}, \rho^{(t)}$ are the corresponding decision-maker output and reward (see eq. 2). Thompson Sampling approaches rely on a flexible Bayesian regressor, which is commonly taken to be a GP. A slight drawback is that GPs have difficulties scaling to larger datasets and must employ approximations and other optimizations (Murphy, 2022). A larger drawback, however, is that GPs often have difficulties modeling higher-dimensional domains (Murphy, 2022). For DISS, the regressor must model in the joint space of instance features, subset indicators (which encompass the power set over $d$ elements), and additional display options. Below, we propose an alternative to TS with a Bayesian model, enabling the use of any regressor, including our mimic-structured regressor (§ 3.2).

**Frequentist Thompson Sampling (FTS) Acquisition** As previously noted, one may use bootstrap trials to simulate Thompson Sampling without a Bayesian model (Russo et al., 2018; Eckles & Kaptein, 2014; Eckles & Kaptein, 2019). Here, we expand on this strategy for DISS with a mimic-structured regressor by drawing analogues to functional posterior draws and means in a way that enables data acquisition with either TS. Bootstrap trials, based on multiple fits to datasets drawn with replacement, measure uncertainty over statistics in a frequentist fashion (DiCiccio & Efron, 1996). First, we note that under a bootstrap trial with a dataset sampled with replacement, $\widetilde{B}^{(t)}$, the fit of a regressor, $f_{\widetilde{B}^{(t)}}$, is analogous

---

**Algorithm 1** Frequentist Thompson Sampling (FTS)

**Require:** supervised dataset $D$, budget $K$, warmup budget $t_{\text{init}}$, ensemble size $C$, batch size $S$.
1: Let $B^{(t_{\text{init}})} = \{(i_l, b^{(l)}, o^{(l)}, \eta^{(l)}, \rho^{(l)})\}_{l=1}^{t_{\text{init}}}$ (eq. 2), where $i_l, b^{(l)}, o^{(l)}$ are drawn randomly.
2: **for** $t = t_{init} + 1$ **to** $K$ **do**
3:   Let $\widetilde{f}(x, b, o)$ be a regressor fit on $\widetilde{B}$ drawn with replacement from $B^{(t-1)}$.
4:   Randomly draw $i_t \sim \operatorname{Unif}\{1, \ldots, N\}$
5:   Let $(b_t, o_t) = \underset{(b,o) \in \{0,1\}^d \times \mathbb{O}}{\arg\max} \widetilde{f}(x^{(i_t)}, b, o)$
6:   Observe $\eta_t = M(x^{(i_t)} \odot b_t, o_t)$, and $\rho_t = r(y^{(i_t)}, \eta_t)$
7:   Update $B^{(t)} = B^{(t-1)} \cup \{(i_t, b_t, o_t, \eta_t, \rho_t)\}$
8: **end for**

---

to a draw from the posterior. Moreover, the posterior mean is then akin to the mean fit over datasets sampled with replacements, $\bar{f} = \mathbb{E}_{\widetilde{B}^{(t)}}[f_{\widetilde{B}^{(t)}}]$, which may be approximated using the mean over $C$ datasets. This enables one to use any regressor, including those that scale computationally and are more adept in high-dimensions (e.g. our mimic-structured regressor, or XGBoost (T. Chen & Guestrin, 2016)). FTS is summarized in Algo. 1.

## 4 EXPERIMENTS

We evaluate our DISS framework across diverse settings using various black-box decision-makers (BDMs) and datasets. Our experiments cover four main applications, *each leading to their own reward environment*: 1) aiding biased decision-makers; 2) selecting among multiple experts; 3) building interpretable predictors; 4) assisting LLM-based decision-makers. We use public datasets including Skin Segmentation ('skin') (R. Bhatt & Dhall, 2009), Statlog Shuttle ('space') (Henery & Taylor, 1992), SUSY (Whiteson, 2014), Dataset for Sensorless Diagnosis ('sensorless') (Bator, 2013), and California Housing (Pace & Ronald Barry, n.d.) ('housing') (see Appx. B). Across applications and datasets, our experiments span *21 distinct environments* using BDMs ranging from

non-parametric Nadaraya-Watson estimators to logistic regression and LLMs. Unless otherwise noted, we use negative cross-entropy loss between the ground truth label and the decision-maker's prediction as the reward signal to maximize.

To demonstrate the capability of our proposed mimic-structured regressor, we compared it against various methodologies. We note that the large actions space (which encompasses the power set over dimensions), makes it challenging to scale most reward maximization approaches. We compare primarily to recent work, Modiste (U. Bhatt et al., 2023), which considers nearest neighbors based UCB (KNN-UCB) (Guan & Jiang, 2018) contextual bandit scheme to aid decision makers. Note, however, the original Modiste paper only considers a few actions (typically three) to customize data fed to decisions. We consider two variants of Modiste: `Modiste-KNN` and `Modiste-UKNN` (see Appx. C for details). For further context, we also used Continuous Multi-task Thompson Sampling Poseterior Mean (`MTSPM`, Char et al., 2019) Thompson Sample (`TS`, Thompson, 1933; Thompson, 1935; Russo et al., 2018), expected information (`EI`, Mockus et al., 1978), regional expected value of improvement (`REVI`, Pearce & Branke, 2018), and random (`Random`) acquisition strategies with our proposed frequentist bootstrap routine that utilizes XGB Regressors (T. Chen & Guestrin, 2016) for better scaling. We additionally compare to SpannerIGW (Zhu et al., 2022), a large action space contextual bandit approach. However, it requires significantly more interactions with the environment (budget) to reach comparable rewards; thus, the results of this comparison (along with further details) are provided in Appx. E.

We plot held-out test time rewards vs number of acquired observations to report policies' inference time performance under various budgets and observe that our proposed method outperforms other baselines we compared to. In addition, we also profiled and observed no significant impact on runtime between our proposed mimic-structured regressor and vanilla bootstrap XGB regressor; see Appx. D for details. Our code will be open-sourced upon publication.

## 4.1 MIMIC-STRUCTURED REGRESSOR ABLATION

We begin by showing that our proposed mimic-structured regressor (`bootstrap-mimic`) yields better performance at providing decision support compared to a vanilla bootstrap XG-Boost regressor (`bootstrap-xgb`) *regardless of the choice of contextual bandit acquisition strategies*. We show results in Fig. 2 for 'simplicity bias' environments (expounded on below in § 4.2); ablations for all other environments can be found in Appx. H, and follow similar trends. These results indicate that our proposed mimic-structured regressor can better utilize the data at hand for decision support of BDMs, and can better expose it's current uncertainties to actively acquire observations (across a myriad of strategies). For experiments in the following sections, we label our mimic-structured model with Thompson sampling as `Mimic` and compare to other baselines as explained previously.

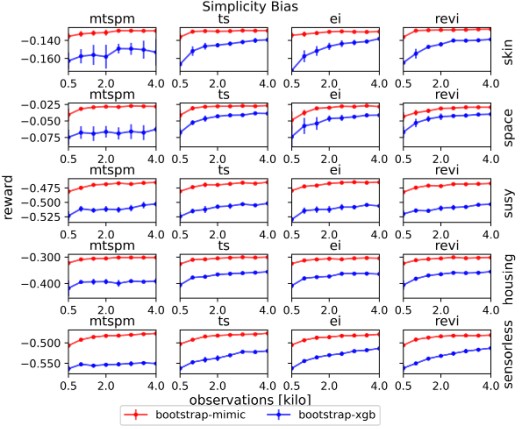

Figure 2: Using the simplicity bias environment, we observe that our structured-mimic approach always outperforms bootstrapping with XGBoost regressors across all acquisition strategies.

## 4.2 BIASED PREDICTOR SUPPORT ENVIRONMENTS

Often, decision makers have limitations and biases that hamper their performance. For example, human cognition has many well-studied limitations (Griffiths, 2020; Buschman et al., 2011; Caruana et al., 2020), including cognitive overload (Iskander, 2018) and simplicity bias (J. Chen et al., 2001; Schulman et al., 1999). Hence, on a per-instance basis, we propose training DISS policies to identify and forward a configuration (an informative feature subset $b$ and an option $o$) to the decision maker for prediction (e.g. classification). The inputs that a decision-maker receives include: 1) a subset of the information on an instance (as indicated by the masked feature vector $x \odot b$); 2) additional

options $o$ (represented in vector form, if applicable, e.g. different ways for visualizing information). We devise two sets of experiments *simulating* cognitive overload (Iskander, 2018) and simplicity bias (J. Chen et al., 2001; Schulman et al., 1999) through a Nadaraya-Watson label smoother; here, Nadaraya-Watson label smoothers adopted in this section of experiment can make predictions using arbitrary feature subsets and outputs its prediction in the form of class probability distributions (see Appx. E.1). Below, we describe each set of experiments in detail.

**Cognitive Overload** Human decision makers are often hampered by the limited ability of their working memory to hold multiple simultaneous pieces of information. When these limits are reached, one is likely to experience "cognitive overload" and therefore have difficulties

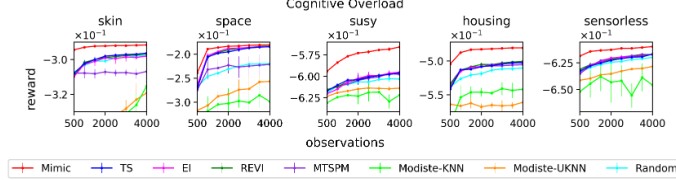

Figure 3: Avg. reward vs. data budget on cognitive overload envs.

making accurate prediction with confidence (Iskander, 2018). We simulate the effects of cognitive overload by applying a temperature-controlled softmax activation function to the label probabilities obtained from our synthetic expert: $\text{softmax}(\hat{\mathbf{y}}, b)_i = \frac{\exp\left(\frac{\hat{y}_i}{T(b)}\right)}{\sum_{j=1}^{\text{nclass}} \exp\left(\frac{\hat{y}_j}{T(b)}\right)}$ where $T(b)$ is a temperature function *whose value increases as more features are observed* in $b$ (we set $T(b)$ to be proportional to the 2-norm of $b$). Note that, for this BDM, certainty decreases when more features are forwarded (simulating cognitive overload). Thus, in order to provide effective decision support to this BDM, we expect our policy to learn a small but efficient feature subset that allows accurate prediction while incurring minimal uncertainty penalties. In Fig. 3, we see that in this setting, our `Mimic` policies handily outperform other baselines; in fact, the rewards attained by `Mimic` after the warm-up period of 500 observations alone commonly exceed the other baselines' highest reported rewards.

**Simplicity Bias** Rather than viewing data profile holistically, humans may allocate undue attention towards prejudicial factors or simple explanations stemming from a few features; e.g. racial and gender biases in healthcare are known to lead to discrepancies in treat-

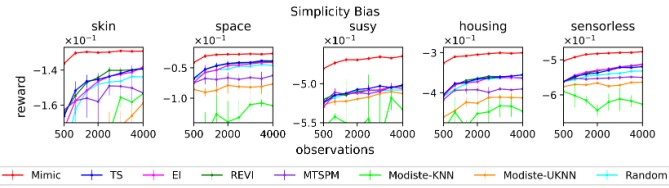

Figure 4: Avg. reward vs. data budget on simplicity bias envs.

ment recommendations (J. Chen et al., 2001; Schulman et al., 1999). We simulate a simplicity bias environment for prediction by computing a combination between the BDM's original prediction and a simplified prediction based on a sole feature. In this case, the biased prediction comes from a univariate classification model $g(x_j)$ trained only on the $j^{\text{th}}$ feature (the "poison feature."). If the poison feature is selected in $b$, the expert's original prediction will be modified as $\widehat{y}_{\text{poison}} = g(x_j)$; making for a BDM that bases its prediction entirely upon the poison feature if it is present. Otherwise, the prediction will be unchanged, and the BDM predicts using all the forwarded features. To avoid the effects of simplicity bias, we expect our method to learn a policy that avoids including the poisoned feature in the subsets presented to the expert (in cases where the feature would yield incorrect decisions). Here, on higher-dimensional datasets ('space,' 'susy,' and 'sensorless'), we observe `Modiste` struggles to consistently increase its rewards as more examples are observed. While policies trained with bootstrap XGB regressor aid performance, we consistently see the best rewards from our proposed `Mimic` structured approach.

## 4.3 MULTIPLE EXPERT ASSIGNMENT ENVIRONMENTS

Next, we explore how one can utilize the additional options $o$ to enable selecting among multiple experts. Decision makers in practice often have varying expertise on different instance sub-spaces; thus, in this setting, DISS policy must also choose which decision-maker to utilize in addition to picking a feature subset to display. We write the BDM as $M(x \odot b, (j, o)) = M_j(x \odot b, o)$, where the option $j$ indexes which of the available decision-makers $(M_1, \ldots M_J)$ to apply to an instance with the subset $b$ and further options $o$. For this experiment, we set up a multi-expert environment

by using K-Means clustering on each dataset with $k = 4$, resulting in four distinct experts (also based on Nadaraya-watson label smoother) for our policy to choose from. Unlike § 4.2, synthetic experts in these experiments were not modified with any additional source of limitation or bias; only the source of the models' training data differed between experts (see Appx. E.2). Results in Fig. 5 shows structure mimic regressor consistently outperforms all other baselines. In fact, rewards earned by the mimic policy reach near the max of 0.0 for two datasets and suggests that options to select among decision makers enables better overall predictions compared to relying on a single expert.

## 4.4 INTERPRETABILITY AND VISUALIZATION ENVIRONMENTS

Interpretability of ML models is hindered when making predictions in higher dimensional domains (Karim et al., 2023). Static feature selection can help interpretability of predictions, especially when utilizing a small number ($\leq 3$) of features, which enables visualization. Unfortunately, a small static feature set is unlikely to yield good predictions in most applications. Thus, we propose utilizing DISS to select a *dynamic* (per-instance) feature subset to utilize for making predictions (which can then be visualized easily). We do so as follows: first, a cardinal-

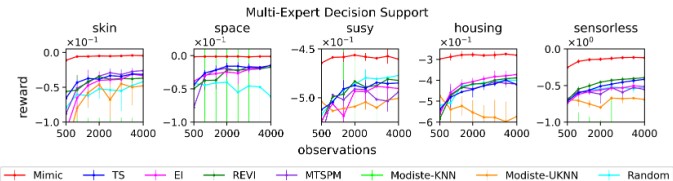

Figure 5: Avg. reward vs. data budget on multi-expert envs.

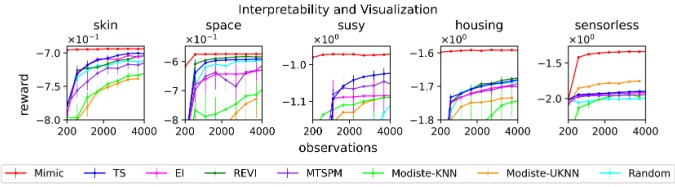

Figure 6: Avg. reward vs. data budget on interp. envs ($\lambda = 0.5$).

ity penalty is added to any reward to promote efficacious decisions with fewer features, i.e. $r_{\text{new}}(y, M(x \odot b, o)) = r(y, M(x \odot b, o)) - \lambda \|b\|_1$; second, we enforce interpretability by explicitly restricting the action space of the DISS policy to only consider subsets of a small dimensionality (e.g. 3: $\{b \mid \|b\| \leq 3\}$). By fitting the classifier to the specific DISS policy selected features on training data, we may utilize this approach with any type of classifier as the BDM, e.g. logistic regression classifier used in this section of experiments. The restriction on the number of potentially selected features per instance enable us to easily visualize the entire dataset (and predictions) on the dimensions selected.

Through Fig. 6, we again see our `Mimic` approach typically reaches top performance faster than all other baselines, verifying our intuition in taking advantage of our prior knowledge of reward structure. From left panel of Fig. 7, we see our `Mimic` policies are adept at dynamically finding a feature subset to predict with, i.e. DISS can use two or fewer features to achieve similar accuracies compare to a model trained to use all features. In the right panel of Fig. 7, using 'skin' dataset, we visualize two

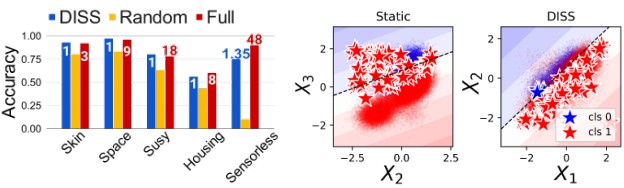

Figure 7: *Left*: Accuracy of `Mimic` vs. random selected features vs. full feature classification (reporting avg. number of features selected). *Right*: Classifier using statically selected features (Static) vs. DISS dynamically chosen feature subset (DISS). Instances to predict and training data are visualized as stars and points, respectively.

linear classifiers trained on two different feature subsets. The DISS classifier plot shows the decision boundary using $x_1$ and $x_2$ on our training data. We show test time points that were selected for this classifier (using $x_1$ and $x_2$ features) as stars. Note that the DISS policy did a good job of mapping test time instances to this subset of features since we see that the star points are well classified with the respective boundary. The `Static` classifier plot shows the decision boundary when training using features $x_2$ and $x_3$ on our training data. Here, the static classifier features ($x_2$ and $x_3$) indicate the best static feature subset as optimized with an feature selection algorithm (w/ ANOVA F-value) as an ML pre-processing step. We see that many of the test points (stars) correctly

classified with the dynamic DISS policy would have been incorrectly classified with the static best policy.

### 4.5 LARGE LANGUAGE MODEL SUPPORT ENVIRONMENTS

LLMs are effective in distilling common-sense knowledge across various domains, making them promising tools for general-purpose decision-making (Yao et al., 2024); Unfortunately, LLMs are sensitive to how input text is structured (Liu et al., 2024; Z. Zhao et al., 2021; Anagnostidis & Bulian, 2024). In this context, we study DISS's ability to enhance decision support for black-box LLMs to make decisions on unseen data. That is, with housing dataset containing interpretable features, and for each test-time instance, we use DISS policy to select a feature subset to include in the question prompt forwarded to the LLM for making a classification; we ask the LLM to return its responses (in natural language) in a way such that we can extract its predictions in the form of categorical distribution, so that we can further compute rewards and evalutate the LLM's level of confidence at making prediction. In this sec-

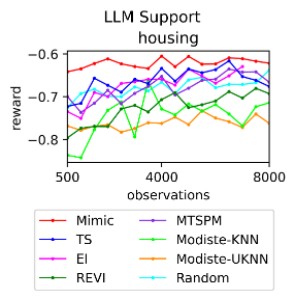

Figure 8: Avg. reward vs. data budget on LLM env.

tion of experiments, we use `Meta-Llama-3.1-8B-Instruct` as our black-box LLM since it achieves strong performance on numerous benchmarks (Dubey et al., 2024), and we asks the LLM to classify whether a real-estate listings is above \$250K using DISS dynamically (on a per-instance basis) selected subset of features. Interestingly, we can achieve better output predictions and confidences by strategically withholding certain information from the LLM (see Appx. F). Again, our proposed `Mimic` approach has the best performance right from the initial 500 observations (see Fig. 8; additional results and details on LLM support through DISS can be found in Appx. E.4

### 4.6 HYPERPARAMETER SENSITIVITY ANALYSIS

We further conducted sensitivity analyses on key hyperparameters of our DISS framework, including: initial capital $t_{\text{init}}$ (warm-up budget), and ensemble size of mimic regressors, $C$, used to estimate the "posterior mean" when deploying the policy (see § 3.2). Due to space constraint, key findings are summarized in this section; complete analysis can be found in Appx. G. Across all tested datasets in the simplicity bias environment, our method demonstrates robustness to hyperparameter choices. For initial capital $t_{\text{init}}$, performance remains stable across values ranging from 10 to 500 warm-up interactions. The ensemble size ($C$) analysis shows consistent performance across ensemble sizes from 2 to 30, indicating that our mimic-structured regressor with frequentist Thompson sampling is largely insensitive to this parameter. Robustness across these hyperparameters suggests that practitioners can deploy DISS without extensive hyperparameter tuning, making it practical for real-world applications where computational budget for hyperparameter optimization may be limited.

## 5 CONCLUSION

We present a novel AI assistance framework, DISS, to enhance the performance of black-box decision-makers by dynamically tuning the presented information. We develop a frequentist approach for DISS policies to acquire data that enables the use of arbitrary regressors; moreover, we propose a novel mimicking approach that takes advantage of the structure of the DISS regression task to better utilize the data at hand. Our experiments provide a broad testbed for decision support in a vast number of environments, ranging from different datasets, with different decision-makers, and different tasks. We show our proposed DISS can support various real-world applications (e.g. biased decision support, expert selection, interpretable modeling, and LLM support) and black-box decision makers (e.g. Nadaraya-Watson estimators, logistic regression, and LLMs). Extensive experimentation (*across 21 total environments*) shows our proposed `Mimic` approach achieves state-of-the-art performance. We believe that the success of our `Mimic` based policies in these diverse experiments can be used as a stepping stone for further development across these applications in future work. For example, DISS may be used for fine-grain dynamic classifier ensembling, or for decision support to mitigate biases, or for efficient inference on resource constrained edge-devices.

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
