# OpenReview forum: "Dynamic Information Sub-Selection for Decision Support"
_ICLR.cc/2026/Conference — ICLR 2026 Conference Withdrawn Submission_

### Official Review · Reviewer_Mrxm · 2025-10-14

**Soundness:** 3
**Presentation:** 4
**Contribution:** 3
**Rating:** 6
**Confidence:** 4

**Summary:**

The paper introduces Dynamic Information Sub-Selection (DISS), a framework designed to enhance the performance of a black-box decision maker (BDM) by learning an instance-specific policy. This policy dynamically determines which subset of features to present and which discrete 'option' (e.g., which expert to query, which prompt template to use) is optimal for the task at hand. The goal is to maximize the efficacy of the BDM's decision, measured by a reward signal, under a limited budget for querying the BDM during training.
The core technical contributions are twofold: (i) a Mimic-Structured Regressor, which insightfully decomposes the complex reward estimation task; and (ii) a scalable Frequentist Thompson Sampling strategy that uses bootstrapping to enable exploration and data acquisition. The framework is extensively evaluated across four distinct application domains and demonstrates significant gains.

**Strengths:**

* **[S1] Novel and Well-Motivated Methodological Core:** The mimic-structured regression elegantly factorizes the learning problem, backed by clear theoretical intuition and proof.

* **[S2] Impressive Generality and Unification:** A key strength is the successful unification of four distinct and important application areas under the single DISS framework. This demonstrates the flexibility and broad relevance of the proposed approach, framing it not as a niche solution but as a general-purpose recipe for optimizing human-AI and AI-AI interactions in resource-constrained settings.

* **[S3] Empirical Rigor:**  Extensive experiments with ablations, full-model baselines, and sensitivity analyses lend high confidence in robustness. Inclusion of timing studies, compute constraints, and hyperparameter stability adds real-world credibility.

**Weaknesses:**

* **[W1] Gap Between Theory and Practice (Adaptivity):** The main theoretical weakness is the mismatch between the assumptions of the risk bound analysis and the practical implementation. The proof in Appendix A assumes that the observations used to train the mimic model $\hat{M}$ are i.i.d. However, the FTS algorithm collects these observations *adaptively*, which breaks the i.i.d. assumption. While the empirical results are strong, bridging this theoretical gap (perhaps by discussing how the analysis might extend to the adaptive setting via martingale concentration arguments) would make the paper's formal claims more robust.

* **[W2] Experimental Realism:** My primary concern remains the external validity of the experimental validation. While methodologically sound, the experiments are conducted almost exclusively on standard tabular UCI datasets with synthetic BDMs. The frequent motivation of *human decision support* is not fully substantiated without human-in-the-loop experiments, which are the gold standard for validating claims about improving human performance. While I understand such studies are resource-intensive, their absence means the paper's claims in this domain remain a compelling but unproven hypothesis.

* **[W3] Practical Significance of Gains:** While statistically consistent, some improvements appear numerically small; clearer discussion of *practical significance* would help contextualize their impact.

**Questions:**

The weaknesses section already incorporate most questions, but I was also concerned about compute requirements and scalability. The experiments were run on a high-end workstation with 512GB of RAM. Could you discuss the memory and time complexity of the approach as a function of the feature dimension, the number of candidate actions, and the budget? Is it feasible to run these experiments on more commodity hardware, or is the memory footprint a potential bottleneck for wider adoption?

---

### Official Review · Reviewer_3TdX · 2025-10-30

**Soundness:** 3
**Presentation:** 3
**Contribution:** 3
**Rating:** 6
**Confidence:** 3

**Summary:**

How can AI systems help dynamically reduce the amount of information a human decision maker needs to incorporate before they make a decision? The authors develop a new framework, Dynamic Information Sub-Selection (DISS), which identifies and filters the amount of information a decision maker needs. The authors apply DISS to a variety of decision making settings.

**Strengths:**

The authors have a strongly motivated piece. I found the paper generally well-written and appreciated the rich and thoughtful connections to people's cognitive constraints. I thought the authors took on quite an ambitious, wide-ranging suite of decision making settings to evaluate across (e.g., I very much enjoyed that the authors directly connected back --- and concretely empirically connected back --- to the points they raised in their "Applications" section on page 2 [something not done in enough works!])

**Weaknesses:**

Perhaps the biggest weakness of the current work is some confusion in the set-up of DISS and Mimic. I found myself needing to reread the text quite a few times to delineate what DISS is relative to Mimic (and what the authors' contributions are therein?)

And while the empirical work is strong, the Modiste baseline (as the authors note) was not really designed to extend to the kinds of tasks the authors considered here wrt the number of possible actions (hence is quite a weak/ill-suited baseline here).

There are also a few related works to information downsampling as it relates to human cognitive constraints that may be worth connecting to in related work: https://openaccess.thecvf.com/content/CVPR2023/html/Ramaswamy_Overlooked_Factors_in_Concept-Based_Explanations_Dataset_Choice_Concept_Learnability_and_CVPR_2023_paper.html and https://ojs.aaai.org/index.php/HCOMP/article/view/27543

I also found the particular motivation of the LLM decision making tasks a bit weak --- what is the value or set-up of the decision task of determining if "the real-estate listings is above $250K"? Ofc I recognize the paper is limited in space and the authors do expand on these analyses in the Appendix; however, in the current state, the task described as in the main text is quite unclear.

**Questions:**

- What are the error bars in the figures? (stdev? standard error? 95% CIs around the mean? something else?)
- Out of scope for this, but a good next work -- given the close links to human cognitive limitations, it would be great to have a user study actually assessing how well DISS/Mimic actually helps people.


Minor:
- Typo in appendix "The following are is an" (in LLM section)

---

### Official Review · Reviewer_PM8r · 2025-10-31

**Soundness:** 2
**Presentation:** 2
**Contribution:** 2
**Rating:** 2
**Confidence:** 4

**Summary:**

- This paper introduces the Dynamic Information Sub Selection method to tailor information shown to a blackbox decision-maker (which can be a human or some other system).
- DISS has multiple potential applications, including prediction support, expert selection, and LLM decision-making.
- The authors compare their mimic approach to a wide range of prior works and baselines, demonstrating the benefits of their Mimic policies based on DISS.

**Strengths:**

- There is potential for broader applicability of this general purpose approach.
- The authors begin to demonstrate generality by the diversity of experimental setups in this work.

**Weaknesses:**

- The Mimic approach assumes the reward function r is known, yet the paper does not justify why this is realistic, especially for human-in-the-loop scenarios.
- The paper tries to address too many applications (e.g., interpretability, human decision-making, and LLM decision support) without properly justifying and motivating each one.
- The experiments rely primarily on standard UCI tabular datasets, where a well-trained model could already achieve high performance, the evaluation set up feels artificial.
- The random baseline performs surprisingly well, sometimes outperforming other methods, yet the paper offers no analysis of why this occurs or what it implies about the experimental design.
- The cross-dataset and cross-baseline analysis is superficial; providing more evidence of the utility of the proposed decomposition would be helpful.

**Questions:**

Please address each of the weaknesses above.

---

### Note · Authors · 2025-12-01

I have read and agree with the venue's withdrawal policy on behalf of myself and my co-authors.